# Black–White Risk Differentials in COVID-19 (SARS-COV2) Transmission, Mortality and Case Fatality in the United States: Translational Epidemiologic Perspective and Challenges

**DOI:** 10.3390/ijerph17124322

**Published:** 2020-06-17

**Authors:** Laurens Holmes, Michael Enwere, Janille Williams, Benjamin Ogundele, Prachi Chavan, Tatiana Piccoli, Chinacherem Chinaka, Camillia Comeaux, Lavisha Pelaez, Osatohamwen Okundaye, Leslie Stalnaker, Fanta Kalle, Keeti Deepika, Glen Philipcien, Maura Poleon, Gbadebo Ogungbade, Hikma Elmi, Valescia John, Kirk W. Dabney

**Affiliations:** 1Nemours Children’s Healthcare System, Wilmington, DE 19803, USA; Michael.Enwere@nemours.org (M.E.); jwilliams2@arthurasheinstitute.org (J.W.); benjaminogundele@gmail.com (B.O.); pchavan3@buffalo.edu (P.C.); Tatiana.Piccoli@nemours.org (T.P.); Chinaka.Chinacherem@nemours.org (C.C.); Camillia.Comeaux@nemours.org (C.C.); lavisha.pelaez@nemours.org (L.P.); Michael.okundaye@phila.gov (O.O.); Leslie.Stalnaker@nemours.org (L.S.); fkalle@udel.edu (F.K.); Keerti.Deepikaallam@nemours.org (K.D.); Hikma.Elmi@nemours.org (H.E.); vxjohn03@gmail.com (V.J.); kirk.dabney@nemours.org (K.W.D.); 2Biological Sciences Department, University of Delaware, Newark, DE 19716, USA; 3Emergency Department, Thomas Jefferson University, College of population Health, Philadelphia, PA 19107, USA; 4Fellow of Translational Health Disparities Science (FTHDS), Wilmington, DE 19803, USA; 5Public Health Department, Walden University, Minneapolis, MN 55401, USA; 6Edward Via College of Osteopathic Medicine, Auburn, AL 36832, USA; 7Emergency Department, Victoria Hospital, Castries, St. Lucia; vh@govt.lc; 8School of Nursing, Florida International University, Miami, FL 33139, USA; maurrap@icloud.com; 9Global Health Services Initiatives Incorporated, Arlington, TX 76014 USA; debogungbade@gmail.com

**Keywords:** COVID-19 (SARS-COV2), race/ethnicity, case fatality, mortality, health disparities, United States

## Abstract

Background: Social and health inequities predispose vulnerable populations to adverse morbidity and mortality outcomes of epidemics and pandemics. While racial disparities in cumulative incidence (CmI) and mortality from the influenza pandemics of 1918 and 2009 implicated Blacks with survival disadvantage relative to Whites in the United States, COVID-19 currently indicates comparable disparities. We aimed to: (a) assess COVID-19 CmI by race, (b) determine the Black–White case fatality (CF) and risk differentials, and (c) apply explanatory model for mortality risk differentials. Methods: COVID-19 data on confirmed cases and deaths by selective states health departments were assessed using a cross-sectional ecologic design. Chi-square was used for CF independence, while binomial regression model for the Black–White risk differentials. Results: The COVID-19 mortality CmI indicated Blacks/AA with 34% of the total mortality in the United States, albeit their 13% population size. The COVID-19 CF was higher among Blacks/AA relative to Whites; Maryland, (2.7% vs. 2.5%), Wisconsin (7.4% vs. 4.8%), Illinois (4.8% vs. 4.2%), Chicago (5.9% vs. 3.2%), Detroit (Michigan), 7.2% and St. John the Baptist Parish (Louisiana), 7.9%. Blacks/AA compared to Whites in Michigan were 15% more likely to die, CmI risk ratio (CmIRR) = 1.15, 95% CI, 1.01–1.32. Blacks/AA relative to Whites in Illinois were 13% more likely to die, CmIRR = 1.13, 95% CI, 0.93–1.39, while Blacks/AA compared to Whites in Wisconsin were 51% more likely to die, CmIRR = 1.51, 95% CI, 1.10–2.10. In Chicago, Blacks/AA were more than twice as likely to die, CmIRR = 2.24, 95% CI, 1.36–3.88. Conclusion: Substantial racial/ethnic disparities are observed in COVID-19 CF and mortality with Blacks/AA disproportionately affected across the United States.

## 1. Introduction

The understanding of the social gradient, implying the social determinants of health and health-related events is essential in transforming health equity in epidemics and pandemics. Health distributions and determinants reflect social inequity, which is the unfair and unjust distribution of social, economic and environment conditions associated with health [1] resulting in health disparities and inequitable outcomes of morbidity and mortality [2,3]. Whereas smoking, physical activities, unhealthy nutrients and excessive alcohol consumption have been linked to health outcomes due to decreased immune responsiveness and abnormal cellular differentiation and maturation [3,4], the understanding of these determinants in infectious diseases, such as COVID-19, is relevant in epidemic curve flattening and case fatality mitigation. Epidemiologic data clearly implicate the sub-populations characterized by low education, low socio-economic status, disadvantaged or deprived neighborhoods with excess morbidity and mortality in any epidemic and pandemic [5,6], relative to socio-economically advantaged [7,8]. Translational epidemiology that involves research at biologic, clinical correlates and population-based levels provides the perspective and challenges in addressing and transforming health equity during pandemics such as COVID-19.

Translational epidemiology requires causal chain model in addressing epidemics and pandemics, which should be based on the social determinants of health (SDH), such as the conditions in environments in which people live, learn, work, worship, age and play. Specifically, these conditions reflect the availability of resources in addressing quality education, economic stability, health and healthcare needs, safe neighborhood and built environment, as well as social and community context [9]. These attributes of SDH reflect the social gradient in health, implying impaired health outcomes for the socially disadvantaged such as low socio-economic status (SES) relative to the socially advantaged (wealth, income, high education). Available data on pandemic history, such as H7N9 implicate poverty, inequity, and SDH in infectious disease transmission and the associated burden of morbidity and mortality [10]. The 2009 H1N1 influenza pandemic observed differential mortalities by socio-demographic [10] as well as substantial geographic variation [11] and population density [12]. In addition, the H1N1 pandemic observed an increased rate of hospitalization among the poor and those residing in poor neighborhoods as well as racial/ethnic minorities in the US [13]. Observed in the 1918–1920 flu pandemic mortality was low SES as per-head income associated with healthcare access, care utilization, nutritional status and comorbidities [14]. A study on the flu pandemic in 1918 in Chicago observed significant disparities in transmission and mortality associated with socio-demographics namely population density, illiteracy, unemployment, and age. Specifically, in relation to illiteracy, the multivariable model after controlling for population density, homeownership, employment and age, showed a 32.2% increased mortality risk for every 10% increase in illiteracy [15].

Blacks/African Americans (AA) as racial minorities are disproportionately affected by almost all pathologies in the US except tuberculosis (TB), cystic fibrosis, suicide, alcoholism and unintentional injuries as in motor vehicular accidents [16]. This population experienced excess adverse outcomes and morbidities, which had been attributed to social inequity and health inequity as exposure function of the SDH [17], psychosocial stressors and unfair and unjust incarcerations. A recent review of race and 1918 influenza pandemic in the USA observed lower transmission (13% vs. 25%) among Blacks but higher case fatality (1–2% vs. 2–3%) relative to Whites [18]. The current COVID-19 global pandemic, caused by SARS-COV2, a ribonucleic acid (RNA) single-stranded enveloped pathogenic microbe remains very virulent, with the potential to be transmitted by asymptomatic individuals or carriers. While population density increases the risk of COVID-19 as observed in New York City, Detroit and Los Angeles, race as a socio-demographic factor may play a role in transmission as well as mortality and higher case fatality among Blacks/AA compared to their White counterparts. Moreover, relevant to increased or differential exposure to SARS-COV2 is low-income households, as reflected in Blacks/AA residing in apartment complexes and crowded neighborhoods in the US. The higher case fatality among Blacks/AA reflects the poorer health outcomes as observed in Cardiovascular Diseases (CVDs), hypertension, malignant neoplasm, diabetes, asthma, bronchopulmonary diseases, such as Chronic Obstructive Pulmonary Disease (COPD), and immunologic disorders [19,20]. These adverse health outcomes of Blacks/AA relative to their White counterparts reflects a lack of insurance, limited access to primary care provider and racial discrimination in the process of care navigation [21,22].

Regarding viral spread, contact with the exposed individual, either symptomatic or asymptomatic as in SARS-COV2 increases the transmission, which explains the rationale for increased transmission and mortality among Blacks/AA who reside in dense population areas with crowded housing and suffer adverse environmental neighborhood factors, such as limited green spaces, recreational facilities, safe playgrounds and transportation systems. Blacks/AA relative to their White counterparts are more likely to use public transportation systems such as transit buses, which carries a higher probability of contact with infected COVID-19 cases, increasing the risk of infectivity among Blacks/AA [23]. Since a respiratory virus such as SARS-COV2 compromises the airways resulting in acute respiratory distress syndrome, previous exposure to environmental pollutants and toxins precipitates poor prognosis. Such would be the case in a population associated with exposure to environmental toxins and pollutants, particular Blacks/AA [24].

With evidence-based data assessment from pandemics and epidemics, reflecting higher case fatality among the socially disadvantaged, such as the poor, underserved and Blacks/AA (racial minority), this study intends to examine the current experience of COVID-19 in recommending urgent equitable preparedness in addressing mortality and case fatality in the United States. With structural or organized racism as the main predisposition of Blacks/AA to excess pandemic mortality, the application of the public health disproportionate universalism that mandates equitable allocation of resources necessary for optimal health and enhanced survival, should be urgently implemented. The assessment of these variables such as income, SES, health insurance, safe neighborhood environment, access and utilization of quality healthcare, and lifestyles such as smoking, vaping, alcohol, drugs, physical inactivity, which are implicitly driven by structural racism, allow for an effective intervention mapping in addressing what is “avoidable” and “unacceptable”, social inequity as an exposure function of health inequities and disparities in morbidity and mortality.

The pathogenicity and virulence of a microbe depends on its ability to bridge the innate and adaptive immune response. The subpopulations with exposure to circumstance, such as food insecurity, psychosocial stressors and chronic morbidity have an increased propensity for viral transmission and infectivity complication, resulting in increased mortality and case fatality. Within the Black/AA communities, these exposures, in addition to the lack of health insurance, render the immune system incompetent through several mechanisms. One such mechanism refers to the propagation of the conserved transcriptional response to adversity (CTRA) gene, resulting in an adverse consequence on interferon system elaboration during viral infection and decreased antibodies such as immunoglobulin G (IgG) synthesis. In addition, patients’ response to therapeutic agents, such as monoclonal antibodies, anti-inflammatory and antiviral drugs, depend on transcriptomes, which serve as receptors to drugs or medications, or a lack thereof, which implies a decreased response to therapeutic agents. This current study aimed to assess the disproportionate burden of COVID-19 on Blacks/AA relative to Whites, explain the root causes, implying the social gradient and propose scientific and data-driven recommendations in narrowing the gap in case fatality between Whites and Blacks/AA. Specifically, we sought to: (a) examine the frequency of confirmed cases in the United States and, by region and states, namely Illinois, Michigan, Louisiana, Wisconsin, Maryland, New York and California, (b) assess the cumulative incidence (CmI) of COVID-19 mortality by race, and (c) determine the case fatality differentials between Blacks/AA and Whites.

## 2. Materials and Methods

This study was proposed to examine the exposure function of race as a socio-demographic factor associated with COVID-19 transmission, mortality and case fatality, with the intent to make recommendations for equitable resources during infectious disease epidemic and pandemic. With this initiative epidemic curve flattening and case fatality mitigation remains the norm in a pandemic in the USA.

### 2.1. Study Design

Data simulation and cross-sectional ecologic design was used to assess aggregate data on COVID-19 in selected states (WI, MI, LU, IL, MD, NC, NJ, NY, CA) by race. This design allows for an assessment on the correlation between the frequency of COVID-19 confirmed cases, mortality, case fatality and race. The data simulation was performed using data from the 2009 H1N1 with signal amplification and specific risk stratification by race. The data on risk modeling were available on the following states namely WI, MI, LU, IL, MD.

### 2.2. Study Population and Data Source

The population of interest were geographic clusters by geographic locale such as states and race. These states were selected based on previous studies on H1N1 influenza pandemic and the observed health disparities, with Blacks/AA representing the subpopulation with highest morbidity and mortality. The data sources utilized were the state department of health that provide COVID-19 cases and mortality by gender, county or location, age and race. However, these data, although reliable, were not very inclusive, since race data were not available for some COVID-19 confirmed cases and deaths.

### 2.3. Variables Ascertainment

Race as the exposure function of laboratory confirmed cases of SARS-COV2 with ribonucleic acid (RNA) sequencing high-throughput process and mortality was self-reported from all tested individuals, positive or negative. The confirmed cases were based on either state or private laboratory tests on the viral RNA antigen. Other variables available for public access included age, gender, hospitalization, recovered cases and location of cases such as county, city, state, etc.

### 2.4. Statistical Analysis

Data simulation involved the use of an immediate risk ratio to assess the 2009 N1H1 2009 influenza pandemic data. This was performed by cumulative incidence risk. The ecologic or aggregate data on COVID-19 from the states’ public access data was used to determine racial variances or differences in confirmed cases and mortality. The case fatality was performed using the formula:COVID-19 deaths/COVID-19 frequency or confirmed cases × 100.(1)

To determine the racial trends in the case fatality, we utilized the percent change by race:recent case fatality (RCF)–previous or baseline case fatality (BCF)/BCF × 100.(2)

The burden of the disease COVID-19 as a ratio of the shared contribution of subpopulations to the overall mortality was estimated using the formula:proportion or percentage of deaths in subpopulation within the domain/the US Census 2020 estimated percentage of the subpopulation.(3)

For example, if the proportion or percentage of death in population A = 46% based on the total number of deaths in the overall population, and the population size of that population (A) = 14%, the COVID-19 burden for that population = 46/14 = 3.3, implying a disproportionate burden of death in population A.

The binomial regression model (BRM) was used to determine the risk of dying from COVID-19 by specific populations in the race category. This model is adequate in predicting the risk of dying from COVID-19 and by race stratification. The BRM is based on the probability of dying in the populations of interest, namely Whites and Blacks/AA, and with Whites as the selected population in the referenced group for the relative risk assessment, as the risk ratio point estimate or magnitude of effect. In this model, we predicted the risk of dying, given the race of the confirmed COVID-19 case. The response or dependent variable ‘y’ (COVID-19 deaths), with each value in the independent variable (race/ethnicity) representing the number of success (k-deaths) observed in ‘m’ trials. Assuming the probability of success (death) = π, and the probability of failure (alive) = 1 − π. We used the link function to relate the risk of dying to a linear combination of the regression variable, including the constant or intercept on the y axis:ln (pr (success (death) = 1/pr (success (death) = 0) = β0 (constant/intercept) + β1(race).(4)

For example, the risk of dying from COVID-19 among Blacks/AA was compared to the risk of dying among Whites for the risk ratio estimation using 2 × 2 tabulation (A/AB)/(D/CD), A = Confirmed cases and COVID-19 mortality among blacks, B = Confirmed cases with recovery/survival among blacks, C = conformed cases and COVID mortality among whites, while D is confirmed cases with recovery/survival among whites. A/AB is the risk in the exposed, Blacks/AA, divided by the risk in the unexposed, Whites, D/CD. In terms of the interpretation, if the two populations are comparable with respect to the mortality risk, the risk ratio (RR) = 1.00, and if the risk is higher among Blacks/AA, the RR is >1.0, but if the risk of COVID-19 is lower among Blacks/AA, the RR is <1.0^25^.

The type I error tolerance was set at 0.05 (95% CI) and all tests were two tailed. The entire analysis was performed using STATA, Version 15.0 (Stata Corp, College Station, TX, USA).

## 3. Results

These data reflect confirmed cases of COVID-19 and mortality by race from selected states where these data are currently available. Although not on the table, the case fatality of SARS-COV2, the causative pathogen in COVID-19 in the United States as per 04/14/20 was 4.11%, with New York, California, Louisiana as epicenters. As per 15th April, while the case fatality was 6.70% globally, the case fatality in the US was 4.94%.

The cumulative Incidence (CmI) of COVID-19 mortality by race in selective US States and regions is illustrated in Figure 1 and Figure 2. The cumulative Incidence of COVID-19 mortality in selected states and regions or cities with data on race reflects a relative increase in mortality of Blacks/AA with respect to population size, and Whites. These states were IL, LU, MI, NY, NC (Figure 1A) and WI (Milwaukee County). While Blacks/AA represent 13% of the population of the US as per US Census projection (2020), overall mortality was 34% (*n* = 19,062), with a 21% excess cumulative incidence of dying as per the 04/09/20 data with a total death of 5700. In the limited data on mortality by race, while Blacks represent 6% of the CA population size, the mortality data indicated 8% (*n* = 21). With the population size of Blacks/AA in LA county (8%), excess mortality was observed by the reported 16 deaths, (17%). Similarly, whereas Blacks/AA represent 16% of the FL population, the mortality from the limited data, *n* = 318 (race/ethnicity data not available, 354–318) was 20% among Blacks, *n* = 63.

In the state of Illinois, Blacks/AA represent an estimated 14% of the total population but the CmI was 46%, *n* = 225. The population of Blacks/AA in Chicago is 29% but the mortality from COVID-19 was 70% (*n* = 132). Blacks/AA comprise 32% of the Louisiana state population but represented 71% (*n* = 495) of the total deaths (*n* = 702). The Black/AA in Michigan constitute 14% of the total population but the mortality was 53% (*n* = 430).

In NY state, Blacks/AA comprised 16% of the total population but illustrated 22% of mortality (*n* = 659). Similarly, while Latino constituted an estimated 19% of the NY state residents, there was a 24% (*n* = 714) mortality CmI. In contrast, whereas the population size of non-Hispanic or non-Latino Whites in NY state is 55%, the CmI was 43% (*n* = 1278) of the total deaths, *n* = 2940, which is different from the total deaths as per 04/9/20 since race and ethnicity data were not available in most cases (7067—2940 cases). In NY city, Blacks/AA represent 24% of the total population, but mortality was 28% (*n* = 428), while Latino constitute an estimated 29% of the population but mortality was 34%.

In North Carolina, Blacks/AA represent 21% of the total population but the CmI of mortality was 38%, *n* = 23. The population size of Blacks/AA in Milwaukee county is 26%, but the CmI was 66% (*n* = 45). Michigan remains the third epicenter for COVID-19 next to New Jersey, with New York as the number one epicenter in the US. The case fatality in MI as per 15 April was 6.66%, while that of Detroit, the leading epicenter with the highest proportion of Blacks/AA was 6.85%. With the case of fatality in the US, estimated at 4.93% during the same period, COVID-19 mortality disproportionately affects Detroit and Blacks/AA in the city. The mortality proportion in St. John the Baptiste Parish, LU as per mid-April was 7.9%. This population is characterized by environmental pollutants as a result of proximity to chemical plants and refineries.

The frequency of deaths adjusted for the population size in the mid-western states is shown in Figure 2. With the available data on race/ethnicity, Blacks/AA represent an estimated 34% of the total US COVID-19 deaths but make up 12.5–13% of the total US population, indicative of the excess burden of this pathogen on this community of color in IL, Chicago, MI, Milwaukee City and Milwaukee County in the state of Wisconsin.

Figure 3 examines the excess ratio of death based on the population size of Blacks/AA. For example, in IL, Blacks/AA represent 14% of the total state population but the COVID-19 mortality was 46% in this population, indicative of the disproportionate share of the burden of this pathogenic microbe in this community of color.

The risk of dying from COVID-19 comparing Blacks/AA to Whites is illustrated in Figure 4. With Whites as the referenced population in this binomial regression model, the COVID-19 mortality risk was higher in the overall states selected and in all the four states and Chicago.

Table 1 demonstrates the case fatality in the state of Maryland, stratified by race. The frequency of confirmed cases and mortality were 2064 and 55, respectively, for Blacks in 04th April, with a case fatality of 2.7%, while the case fatality for White was 2.5%, χ^2^ (4) =13.6, *p* = 0.001. A follow up period by 4th April indicated increased case fatality among Blacks/AA, 3.3%, resulting in a positive trend in case fatality with respect to period change 22.2%.

Table 2 demonstrates the case fatality in Illinois, Chicago, Michigan and Detroit. In Illinois, the frequency of confirmed cases among Blacks in 9th April was 4209, with 200 deaths resulting in case fatality of 4.8%, while a case fatality among Whites was 4.2%. In Chicago during the 9th April, the case fatality among Blacks/AA was 4.5%, while among Whites it was 2.0%, χ^2^(2) = 52.6, *p* < 0.001. On 13th April there was an increase in case fatality in both races, with Blacks/AA bearing the greater burden of mortality. The case fatality in Blacks/AA was 5.9% and 3.2% for Whites, χ^2^(2) = 110.8, *p* < 0.001. The period percent change (PPC) for Blacks/AA was 31.0% and for Whites, 59.4%, indicative of a positive trend in mortality by these races. A racial variance was observed in CmI and case fatality in Michigan state. The case fatality for Black was 7.4%, while 8.0% was observed among Whites, χ^2^(2) = 126.8, *p* < 0.001. Although specific data are unavailable during this preliminary study, Detroit in Michigan illustrates case fatality disadvantage for Blacks/AA, with Blacks/AA relative to Whites observed to be 5.5 times as likely to die from COVID-19.

Although not illustrated in the table, the Illinois state had a total number of confirmed cases, *n* = 15,078, mortality, *n* = 462, with a case fatality of 3.06% on Wednesday, (4/8/20), while, on Thursday (9th April) there was a case fatality of 3.22% (case, *n* = 16,422, deaths, *n* = 528). As per US census projection, 2020, Blacks/AA comprise 14.6% of the Illinois population, but represent 27.9% and 43.2% of COVID-19 confirmed cases and mortality, respectively. The Wisconsin data on 13th April indicate the total confirmed cases, *n* = 3428, female *n* = 1817 (53%) and male *n* = 1611 (47%), mortality: male *n* = 92 (60%) and female *n* = 62 (40%). The Milwaukee County cases as per 13th April was *n* = 1743, and 94 deaths. In Wisconsin, Blacks represent an estimated 6.7% of the population by the 2019 US census estimate but constituted 25% (*n* = 862) of all the confirmed cases and 42% (*n* = 64) of all deaths.

Table 3 demonstrates the Black–White risk differentials in case fatality with binomial regression models in MI, MD, IL, WI and the city of Chicago. The cumulative incidence risk of dying reflects the exposure function of race in COVID-19 mortality in these states and Chicago. There was a survival disadvantage for Blacks/AA with COVID-19 during the month of April 2020 in Michigan, with a significant 15% increased risk of dying relative to Whites, CmI risk ratio (RR) = 1.15, 95% CI, 1.01–1.32. Relative to Whites in Maryland, there was a 5% increased risk of dying from COVID-19 among Black residents compared to Whites, RR = 1.05, 95% CI, 0.70–1.39. The state of Illinois observed a 13% increased risk of dying for Blacks/AA compared to Whites, RR, 1.13, 95% CI, 0.93–1.39. A 51% increased risk of dying was observed among Blacks relative to Whites in the state of Wisconsin, with most data from Milwaukee city and Milwaukee County, RR = 1.51, 95% CI, 1.10–2.10. In the city of Chicago, Blacks/AA were more than two times as likely to die from COVID-19 relative to Whites, implying that for everyone White death from COVID-19, more than two Blacks/AA will die, RR = 2.24, 95% CI, 1.35–3.88.

## 4. Discussion

With the current COVID-19 pandemic, and the USA with the highest confirmed cases and mortality globally as well as the disproportionately observed mortality in some subpopulations, there remains an urgent need to examine these data, provide a possible explanation to the observed racial disparities and propose feasible recommendations in racial gap narrowing in COVID-19 mortality. Aggregate data were utilized from the various state departments of health, where demographic information was available to determine the racial variances in the confirmed cases and mortality as well as risk differentials, comparing the Whites to the Black/AA subpopulation deaths. This preliminary study provides a few relevant findings to epidemic curve flattening and case fatality mitigation in the communities of color, namely Blacks/AA. First, of the limited data on race examined in the selected states and cities with race information, Blacks/AA despite the population size of 13% in the US, represented 34% of all deaths in the referenced states and cities. Secondly, there was a disproportionate burden of COVID-19 mortality among Blacks/AA. Thirdly, the case fatality was higher among Blacks/AA relative to Whites. Fourthly, Blacks/AA relative to Whites in all states and cities assessed for COVID-19 mortality had a higher risk of dying, implying excess mortality.

This study has demonstrated that, despite the 13% population of Blacks/AA in the US, this population represented an estimated 34% of all deaths from COVID-19 in mid-April 2020. The observed population variance and the highest mortality among Blacks/AA had been observed in recent media reporting [23,25,26,27]. The spread of pathogenic microbes such as viruses historically and to date has been linked to social inequity, where the socially disadvantaged populations convey the highest burden of the disease, including poor prognosis and mortality. The flu or influenza pandemic of the 1918 through 1919 in the US observed a lower cumulative incidence of the disease among Blacks but higher mortality relative to Whites [18]. The current COVID-19 case confirmation among Blacks/AA mirrors the observed lower cases or morbidity among Blacks, due to decreased effort by the US public health system, local and county health departments and the healthcare institutions to provide the assistance needed such as transportation for Blacks/AA to access the screening centers and sites.

Secondly, Blacks/AA are less likely to be screened due to implicit bias [28] and lack of healthcare resources, including private insurance through well-paying jobs relative to Whites [21,22,23]. Clearly, the dynamics of viral pathogens reflects the penetrance in populations with compromised immune systems, driven by structural poverty and food insecurity, which Blacks/AA are more predisposed to compare to Whites. With non-restricted conditions and the relaxation of clinical guidelines in infectious disease diagnosis during the pandemics, more cases are identified, isolated and contact tracing is performed for more screening. This effort lowers the transmission and the spread of the virus, enhances case detection, improves the prognosis and reduces case fatality at a population and individual levels. Such initiative will reduce the unequal burden of morbidity and mortality now and in future pandemics.

Thirdly, illiteracy rate and a lower level of education, which is higher in Blacks compared to Whites due to structured racism and racial residential segregation that deprived Blacks/AA of the opportunity of early childhood education, and quality early education had been observed to have an adverse effect on the transmission and mortality from influenza virus during the 1918 pandemic in Chicago [15]. Since Blacks/AA, historically had been perceived as a vulnerable population in pandemics and in COVID-19, the public health and the affluent societal identification of COVID-19 burden and engagement with this subpopulation of the US society via national public health initiative, will enhance a cooperative effort in data gathering and the understanding of the impact of a pandemic on the overall wellbeing of the Black/AA populations across the nation.

The disproportionate burden of COVID-19 among Blacks/AA had been demonstrated in these findings. Besides this, SARS-COV-2 pathogen in COVID-19, previous pandemics had observed similar burden as in the flu pandemic of 1918 [14,15] and that of 2009 H1N1 influenza pandemic. The observed disparities and the disproportionate burden among Blacks/AA is explained in part by neighborhood-level social determinants, such as unemployment, educational level, housing and living conditions and population density [7,10,29,30]. The epidemiologic modeling of these factors provides reliable data in modulating these effects in anticipation of another pandemic, such as COVID-19-2. The social determinants of health as an exposure function of social and health inequities has been shown in the disproportionate burden in infant mortality, cancer, cardiovascular disease, stroke and diabetes among Blacks/AA. These differentials, where Blacks/AA are at a higher risk of diseases, reflect the unfair, unjust and inequitable distribution or allocation of social, economic and environmental conditions related to health. Additionally, the disproportionate burden of disease as observed among the socially disadvantaged populations, namely Blacks/AA and Hispanics, reflects structural racism, which is different from interpersonal racism.

Racial residential segregation, the geographic clustering of racial and ethnic groups, has been observed to predispose subpopulations to health disparities [31] as observed in this study. Black neighborhoods in Milwaukee, New Orleans (LU), Houston (TX), Brooklyn (NY) Detroit (MI), and South Chicago (IL) are densely populated within apartment buildings. The spread of pathogenic microbes is enhanced by population density and close proximity as well as decreased hand hygiene and environmental conditions, such as contaminants. For example, St. John the Baptiste community in New Orleans, a Black/AA neighborhood, characterized by long-term exposure to environmental pollutants (chemical plants and refineries) represented one of the geographic locations with the highest case fatality in the US, 7.90% as per mid-April 2020. The long-term exposure to air pollutants have an adverse effect on bronchopulmonary function, resulting in chronic respiratory diseases, such as asthma and COPD. Specifically, the accumulation of air pollutants provokes the immune system response through innate and adaptive mechanisms. However, with persistent insult to the immune system adaptation, the host immune system remains incompetent with respect to macrophage activation and neutrophil migration, CD4 cells’ activation and decreased cytokine elaboration, given macrophage inability to present the processed antigen or viral pathogen to CD4 cells for activation and B cells or immunoglobulin growth factors’ development and maturation. With COVID-19 implication in severe respiratory symptoms, populations exposed to long-term air pollutants are expected to present with poorer prognosis and excess mortality from this pathogenic microbe. However, the attributable fraction of exposed (AFE), given the environmental air pollutants in Blacks/AA neighborhoods observed in St. John the Baptist community in LU, remains undetermined, pending the availability of sufficient data in risk estimation.

Additionally, the observed burden of COVID-19 morbidity and mortality in Black/AA communities is partially explained by workplace segregation, where Blacks/AA are employed in jobs with fewer or no benefits, but with more adverse conditions, such as buildings with no fire exit [31,32], as well as a stressful environment. Specifically, Blacks/AA who are employed as public transit drivers have an increased risk of contracting viral microbes, which also explains the COVID-19 racial burden differentials.

This study also observed higher case fatality among Blacks/AA relative to Whites. Previous studies in the flu pandemic of 1918 and 2009 clearly implicated the socially disadvantaged individuals and populations in survival disadvantage following infectivity [18]. The case fatality is a *proportion of those who died as a result of an exposure, divided by all the exposed cohort as the population at risk for mortality, multiplied by 100* in a proportion and not a rate, and reflects viral symptom severity and poor prognosis from COVID-19. The excess case fatality among Blacks/AA is explained in part by the higher prevalence of chronic diseases, namely hypertension and other cardiovascular diseases, diabetes and cancer, as well as aberrant epigenomic modulation in COVID-19 prognosis and mortality. AA/Blacks are more likely, compared to their White counterparts, to be diagnosed with type II diabetes, primary or essential hypertension, stroke and cancer [16] and have higher mortality from these conditions [16]. The higher incidence of these conditions among Blacks/AA had been associated with a lack of access to healthcare as well as decreased healthcare utilization due to several obstacles and barriers [19]. Secondly, the social determinants of health which reflect the needed resources to benefit from optimal health is not equitably available for Blacks/AA. These determinants are characterized by the social gradient upon which the socially disadvantaged individuals or populations are less likely to benefit from early education, quality education through college and a good paying job. The social determinants of health, namely education, socio-economic status, income, employment, food security, racism, safe environment, insurance, transportation, and living conditions, adversely impacts the outcome of morbidity and mortality in Black/AA populations in the US [33].

Since chronic health conditions render the biologic system vulnerable to microbes by compromising the immune system responsiveness, the observed higher case fatality among Blacks/AA is explained in part by the relatively higher prevalence of comorbidities among Blacks relative to Whites. While studies have indicated genetic polymorphism in the predisposition to hypertension as well as epigenetics and epigenomic modulations [34], not much had been considered regarding the interaction of single-nucleotide polymorphism (SNP), epigenomics and the hypertension (HTN) genes involved in immune system regulation. Available genetic and epigenetic studies have implicated hypertension and cardiovascular disease candidate genes in immune system regulation, such as natriuretic peptide B(NAPB1), nuclear factor of activated T cell (NFAT5), Human jagged 1 (JAG1), C-terminal Src Kinase (CKS), Lymphocyte-specific protein 1 (LSP1), adrenomedullin (ADM), auxin-binding protein 1 (ABP1), Feline sarcoma (FES) protein-tyrosine kinase (FES), angiotensin (AGT), serum/glucocorticoid-regulated kinase 1(SGK1), adrenoceptor beta 1 (ADRB1), nerve growth factor receptor (NGFR), cytochrome P450 family 1 subfamily A, member 1 and member 2 (CYP1A1, CYP1A2) and microtubule-associated protein 4 (MAPB4) [35]. These genes mainly serve as negative regulators of B-cells (ABP1), suppress T-cell migration and activation (LSP1), reduce T helper cells activities (ADM), down regulate innate and adaptive immunity (FES) and suppress T cell proliferation. 

The poorer prognosis and excess mortality of Blacks/AA with COVID-19 is in part explained by the highest prevalence of primary HTN and other cardiovascular diseases in this population, due to the candidate genes involved in immune system regulation that are implicated in HTN. In addition, it is not merely the gene candidates but their interaction with the environment that results in transcription factors or transcriptome dysregulation, which is necessary for immunomodulation. Addressing these health inequities will require the public health utilization of a disproportionate universalism approach where the population with healthcare needs are provided equitable resources in transforming health equity in the nation by environmental modification (toxic and air polluted neighborhood, poverty, low income, illiteracy, lower education, violence, incarcerations, etc.).

Because Blacks are predisposed to more environmental pollutants and toxins, psychosocial stressors, a dangerous job environment and incarceration, these environments interact with the gene, implying impaired gene expression and increased disease development, poorer prognosis, increased mortality and survival disadvantage. In addition, because social gradient reflects environmental neighborhood characteristics, the understanding of gene and environment interaction, such as living conditions, may provide an additional strategic approach in intervention mapping for disease management and prevention. In effect, examining the gene and environment interaction, observed as epigenomics, will provide substantial data on intervention in narrowing the gaps between Blacks and Whites with respect to COVID-19 mortality.

Epigenomic modulations that commence at gametogenesis are transgenerational but reversible. The social signal transduction that is evoked from the stress placed on Blacks/AA has a substantial effect on the sympathetic nervous system and provokes the beta-adrenergic receptors. This response has been shown to involve the Conserved Transcriptional Response to Adversity (CTRA) gene expression and the consequent elaboration of pro-inflammatory cytokine due to the impaired gene expression of the transcription factors and the inhibition of gene expression with respect to anti-inflammatory response [36]. In understanding these pathways of genomic stability and their role in disease causation as well as mortality, epigenomic studies are necessary in determining whether or not Black individuals, relative to White individuals, have an increased mean deoxyribonucleic acid (DNA) methylation index with respect to the genome-wide analysis. Such initiative will involve the utilization of the bisulfite pyrosequencing that is very specific in differentiating between the methyl group and hydroxyethyl group, as well as the binding of these groups to the Cytosine-phosphate-Guanine (CpG) region of the gene, inhibiting transcription and the messenger ribonucleic acid (mRNA) sequencing, leading to impaired gene expression and abnormal cellular functionality. The reference to epigenomics investigation reflects the inability of a COVID-19 case to respond to treatment modalities due to the drug receptors in-availability, resulting from impaired gene expression (mRNA translation dysregulation) [37]. The observed epigenomic aberration clearly illustrates treatment effect heterogeneity in which some subpopulations respond differentially to a given therapeutic agent in the phase of epigenomic lesion, explaining in part racial risk differentials in COVID-19 case fatality.

We have illustrated that in the states with data on race, Blacks/AA, relative to Whites, had an increased risk of mortality from COVID-19. The observed higher risk observed in mortality among Blacks/AA, diagnosed with COVID-19 and treated for the disease had been observed in other pandemics, namely the influenza flu of 1918 and 2009 N1H1 [18]. With respect to the geographic locations such as states and cities, the risk ranged from 5% to more than twice as likely compared to Whites (CmIRR, 1.05–2.24). The observed excess risk of dying among Blacks/AA may be explained by the late detection of SARS-COV-2, implying poor prognosis, decreased access to identification and isolation resources, a lack of healthcare resources, and poor-quality care, where Blacks/AA patients are treated with implicit as well as clinician bias [20,25,37]. Also implicated in this excess risk is the social determinants of health as previously observed, where the socially disadvantaged individuals or populations are more predisposed to adverse health outcomes.

Despite the appropriate epidemiologic model used in this preliminary study to determine the Black–White mortality risk differentials in COVID-19, there are some limitations. First, this study used the only available data at this point, which is the aggregate data from the departments of public health, which has a tendency for ecological fallacies. Secondly, there is a tendency for confoundability, since data were not available to assess and control for the confounding in the risk estimation. Thirdly, despite the sources of these data, complete race/ethnicity data were not available in these states and cities utilized in this risk estimation. However, the interpretation of these data remains accurate since an estimated >80% of the race/ethnicity data were available for this modeling. This preliminary study strongly recommends the collection and availability of race/ethnicity data from all states and US territories for the understanding of COVID-19 population dynamics for racial/ethnic minority populations’ engagement and preparedness in the future, for health equity transformation in epidemics and pandemics.

## 5. Conclusions

In summary, racial disparities in COVID-19 case fatality exist and persisted and will continue to persist unless data-driven scientific measures are implemented for racial/ethnic gap narrowing, thus reducing the probability and likelihood of COVID-19-2. Additionally, Blacks/AA have a disproportionate burden and increased risk of COVID-19 mortality compared to their White counterparts. The observed racial disparities is explained by social determinants of health, where Black individuals are exposed to environmental pollutants, social inequity, structural racism, food insecurity, poor housing and living conditions, illiteracy, low SES, and a lack of healthcare resources.

### Recommendations and Future Directions in Pandemic Health Equity Transformation

As part of the strategy in health disparity marginalization in future pandemics in avoiding the emergence of COVID-19-2, this study recommends:

The USA, through the Affordable Care Act, to provide quality healthcare to all Americans, especially Blacks/AA and Hispanics. Such a recommendation, if implemented and evaluated carefully, will reduce the anticipated excess malignant neoplasm (absence of cancer screening and preventive measure due to the pandemic), sleep disorders (sleep pattern alteration due to prolonged REM and shortened non-REM), metabolic syndromes and CVDs (aberrant epigenomic modulation of genes/transcriptome due to increased stressors associated with COVID-19) in populations of color, mainly Blacks/AA and Hispanics.

Secondly, social and physical distancing and utilization of adequate PPE in the healthcare setting until the epidemic curve is flattened in all states and territories prior to return to normal economic life, will prevent a significant resurgence of COVID-19 with altered antigenicity or different serotype (COVID-19-2).

Facemask application throughout the nation, especially among COVID-19 positives and symptomatic individuals to marginalize the spread of the virus.

The immune system potentiation by providing equitable resources to Blacks/AA and Hispanics who are socially disadvantaged with low SES.

The education of the Black/AA communities on the health consequence of COVID-19 and the provision of assistance and encouragement for testing, case identification and isolation will flatten the epidemic curve in the Black communities influencing epidemic curve flattening and down drifting in the epidemic curve nationally, thus enhancing the overall US economy and return to normal life style and coping. This initiative will not only reduce social inequities but the social inequities burden for future disease and pandemics.

Since the disproportionate burden of pandemics fundamentally reflects health disparities, addressing this disproportionate burden in future epidemics and pandemics will require subpopulations, especially blacks/AA, American Indians/Alaska Native and Hispanics to be provided with the resources necessary for optimal health. Therefore, adherence to the World Health Organization’s recommendation of social justice and peace as conditions necessary for health is essential in narrowing health disparities in pandemics by addressing now social injustice and systemic and structural racism, thus transforming health equity.

The federal government through the Center for Disease Control and Prevention (CDC) and the state health departments to increase testing throughout the nation, especially in the most vulnerable COVID-19 population, namely Blacks/AA.

The federal state and local/city government engagement with Black/AA community leaders in assessing the individual and group risk for spread reduction, epidemic curve flattening and case fatality stabilization and mitigation.

The development of a surveillance and monitoring system for data availability on the social determinants on health and race/ethnicity in transforming pandemic health equity.

Since public health is the collective effort by society for all to remain healthy through disease control, prevention and health promotion practices, academic institutions of public health and the healthcare system should advocate professional inclusion and diversity in faculty positions and healthcare provision. This perspective ensures the education of the communities of color on pandemic risk factors in the first place and protective factors by those who are demographically comparable to them, and the provision of care with marginalized implicit and clinician bias to racial and ethnic minorities.

Establishment and implementation of a rapid health equity transformation taskforce and evaluation matrix for risk mitigation among racial and ethnic minorities, namely Blacks/AA and Hispanics.

The federal, state, county and local health departments collect socio demographic data for a better understanding of exposures and confounding in viral spread and case fatality.

As we anticipate a vaccine against COVID-19, the administration of this vaccine must be initiated among the most vulnerable populations, especially the socially disadvantaged population, namely Blacks/AA and Hispanics for herd immunity in these subpopulations prior to other communities’ immunizations.

As emergency preparedness, health equity resources are required through disproportionate universalism, which is the public health lens in the provision of health services more to the racial/ethnic minorities prior to the non-vulnerable populations. The failure to address the socially disadvantaged individuals and populations’ needs in our society, by educational opportunity, equitable employment opportunity and the departure from structural racism, will render the US population more vulnerable to public health and healthcare crises in COVID-19-2. In effect, as clinicians, researchers, health officers, epidemiologist, infectious disease specialist, and public health experts, it is our moral responsibility regardless of our race/ethnicity, gender or age, to rapidly respond to the disproportionate mortality burden of this pandemic in the communities of color, especially Blacks/AA, since human life remains a primary value.

## Figures and Tables

**Figure 1 ijerph-17-04322-f001:**
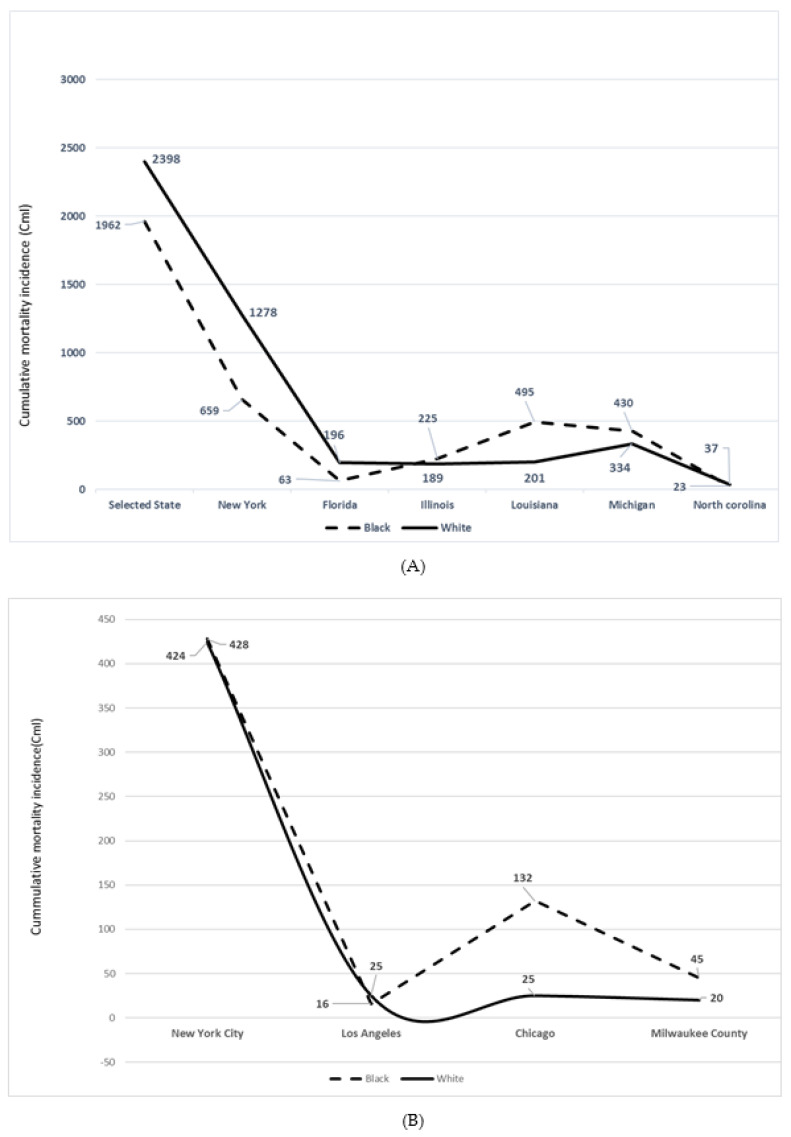
(**A**) Frequency of COVID-19 mortality by race in selected states, USA, 9th April 2020. Notes. The COVID-19 mortality frequency reflects higher occurrence in Illinois, Louisiana, Michigan and North Carolina. (**B**) The frequency of COVID-19 mortality by race in selected cities and Milwaukee County, USA, 9th April 2020. The cities with the disproportionate burden of COVID-19 for Blacks are Chicago and Milwaukee city as well as the county.

**Figure 2 ijerph-17-04322-f002:**
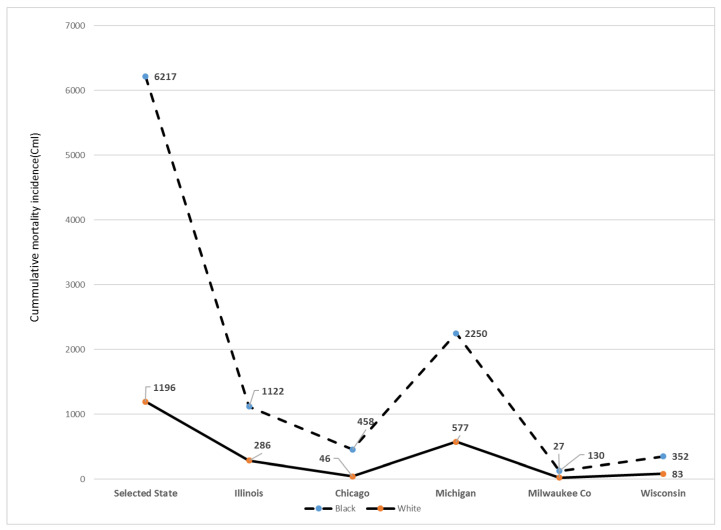
Cumulative incidence of COVID-19 mortality in mid-west states, USA, 9th April 2020. Co = county; in the mid-west states with data on race, mortality cumulative incidence (CmI) was higher among Blacks/AA relative to Whites.

**Figure 3 ijerph-17-04322-f003:**
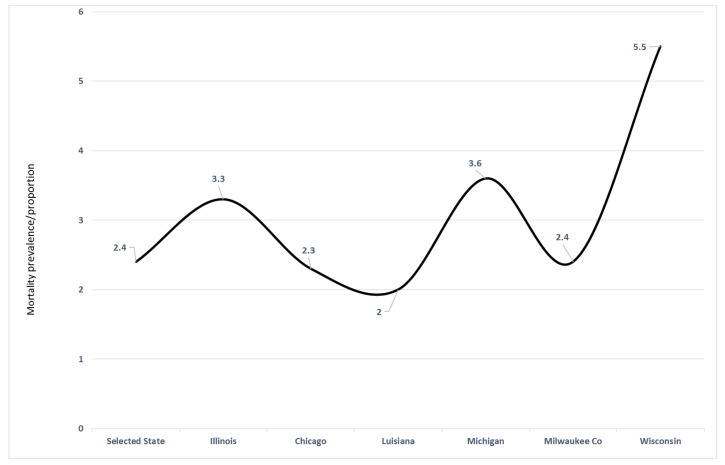
Disproportionate magnitude of COVID-19 mortality on Blacks/AA in selective states and county, 9–13th April 2020. Notes: The point estimates on the line indicates the burden of COVID-19 mortality in the Black/AA communities in these geographic locales. As per mid-April 2020, the COVID-19 disproportionate burden of dying was outstanding in Wisconsin, Michigan and Illinois (mid-western states).

**Figure 4 ijerph-17-04322-f004:**
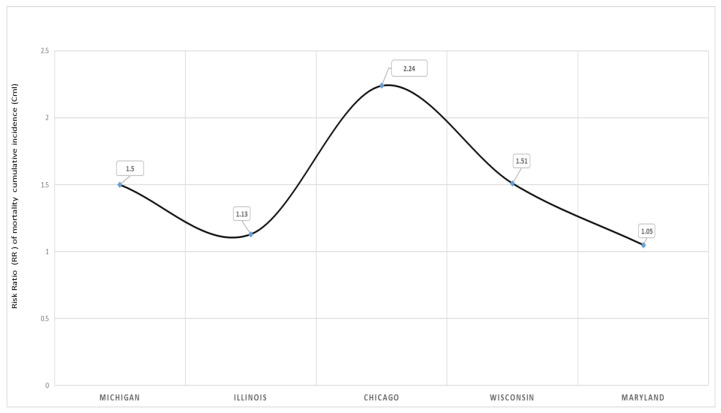
Black–White risk differentials in COVID-19 mortality in selected states and regions in the United States, April 2020. Notes: The solid line indicates the risk of dying from COVID-19 comparing Black/AA with their White counterparts. With Whites as the reference group, implying 1.0 as no risk. In Chicago with a risk ratio (RR) of dying for Blacks/AA estimated at 2.24 in this model, implying that for everyone death from COVID-19 among Whites in Chicago, more than 2 Blacks/AA experienced mortality from the COVID-19 pandemic.

**Table 1 ijerph-17-04322-t001:** COVID-19 case fatality in Maryland by race, the 9th through to the 13th April 2020.

Race	Confirmed Cases	Frequency of Deaths	Fatality Proportion (%)	χ^2^(df), *p*-Value
Maryland ^1^				13.6(4), <0.001
Black/AA	2064	55	2.66	
White	1540	39	2.53	
Asian	122	6	4.92	
Other	449	3	0.67	
Unknown	1354	21	1.55	
Maryland ^2^				21.5(4), <0.001
Black/AA	3202	104	3.25	
White	2305	83	3.60	
Asian	190	8	4.21	
Other	861	12	1.39	
Unknown	1667	28	1.68	

Data Source, Abbreviations and Notes: Maryland Health Department, MD. AA = African Americans, χ^2^ = chi square statistic, df = degree of freedom. 1 = Initial data on April 9th, while 2 = data on April 13th. The type I error tolerance, *p*–value was set at 5% (0.05).

**Table 2 ijerph-17-04322-t002:** COVID-19 case fatality in Illinois, Chicago, Wisconsin by race, the 9th through to the 13th April 2020.

Race	Confirmed Cases (*n*)	Frequency of Deaths (*n*)	Fatality Proportion (%)	χ^2^(df), *p*
Illinois				110.2(2), <0.001
Black/AA	4207	200	4.75	
White	4002	167	4.17	
Unknown	3987	32	0.80	
Chicago^1^				52.6(2), <0.001
White	941	19	2.02	
Black/AA	2102	95	4.52	
Unknown	1450	7	0.48	
Chicago^2^				110.8(2), <0.001
White	1209	39	3.22	
Black/AA	3005	178	5.92	
Unknown	2540	15	0.59	
Wisconsin *				42.0(2), <0.001
White	1726	83	4.81	
Black/AA	862	64	7.43	
Unknown	648	1	0.15	
Michigan				126.8(2), <0.001
Black/AA	8460	625	7.39	
White	6922	577	8.05	
Unknown	7947	305	3.84	

1 = Initial data on April 9th, while 2 = data on April 13th, 2020. χ^2^ = chi square, df = degree of freedom, *p* = probability value for the random error quantification which was set at 5% (0.05) type 1 error tolerance. * Wisconsin reflects the assessment of the COVID-19 case fatality and mortality data from the Milwaukee City and County.

**Table 3 ijerph-17-04322-t003:** Black–White Risk Differentials in Mortality in Selected States, MD, MI, IL, the 9th through to the 13th of April 2020.

State/Race	CmIRR	95% CI	*p*	EAF (%,95% CI)	PAF (%)
Michigan					
White	1.00	referent	referent		
Black/AA	1.15	1.01–1.32	0.04		
Maryland					
White	1.00	referent	referent	5.0(−42–36.5)	2.8
Black	1.05	0.70–1.58	0.81		
Illinois					
White	1.00	referent	referent		
Black/AA	1.13	0.93–1.39	0.22	11.7(−1.0–28)	6.0
City/Race					
Chicago^1^					
White	1.00	referent	referent		
Black/AA	2.24	1.36–3.88	0.001	55.3(25.5–71.)	46.1
Chicago^2^					
White	1.00	referent	referent		
Black/AA	1.79	1.27–2.51	0.001	44.0(21.4–60.)	36.0
Wisconsin					
White	1.00	referent	referent		
Black	1.51	1.10–2.10	0.01	34.0 (9.0–52.0)	15.0

PAF = Population attributable fraction, EAF = Exposure attributable fraction, AA = African American, *p* = probability value for the random error quantification which was set at 5% (0.05) type 1 error tolerance. CmIRR = cumulative incidence risk ratio. 1 = Initial data on April 9th, while 2 = data on 13th April 2020.

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
