# Peer review of "Black–White Risk Differentials in COVID-19 (SARS-COV2) Transmission, Mortality and Case Fatality in the United States: Translational Epidemiologic Perspective and Challenges"

_ijerph, 2020, doi:10.3390/ijerph17124322_

Round 1
Reviewer 1 Report
Please see attachment.

Author Response
Dear Editorial Board Staff
Thanks for the opportunity to revise our manuscript for a reconsideration for publication for your journal. We have addressed these minor comments form the reviewer.
Best regards,
Lead Author
Reviewer 2 Report
In this article, Holmes et al. reviews the social and health inequalities that influence the adverse morbidity and mortality outcomes of COVID-19 pandemics in USA populations. Here they also discuss how the racial and ethnic disparities in deaths associated with pandemic COVID-19 infections in the USA, focusing on disproportionate burden and increased risk of COVID-19 mortality among African Americans [Black/African Americans (AA)] compared to White Americans. The observed racial disparities is explained by social determinants of health, whereby Black are more likely to be exposed to environmental pollutants, social inequity, structural racism, food insecurity, poor housing and living conditions, illiteracy, low SES, and lack of healthcare resources.
The article is well written and the references are cited properly. However, there are some concerns that need to be addressed.
Comments:
- Did the authors observe any sex differences between the Black /AA and their White counterparts in their susceptibility to COVID-19 infections/complications?
- Authors should explain the reasons for choosing the states in this study.
- Authors should provide the percentage of populations for Black /AA and their White counterparts in the states they have studied for the COVID-19 pandemic.
- Authors could simplify figure 1, then it will be easier to understand the COVID-19 mortality graphs.
- In table 2, the authors should consider changing the order of Black /AA and White for all the states; the order should be presented uniformly.

Author Response
Reviewer 2
In this article, Holmes et al. reviews the social and health inequalities that influence the adverse morbidity and mortality outcomes of COVID-19 pandemics in USA populations. Here they also discuss how the racial and ethnic disparities in deaths associated with pandemic COVID-19 infections in the USA, focusing on disproportionate burden and increased risk of COVID-19 mortality among African Americans [Black/African Americans (AA)] compared to White Americans. The observed racial disparities is explained by social determinants of health, whereby Black are more likely to be exposed to environmental pollutants, social inequity, structural racism, food insecurity, poor housing and living conditions, illiteracy, low SES, and lack of healthcare resources.
The article is well written and the references are cited properly. However, there are some concerns that need to be addressed.
Authors Response: Thanks , quite appreciate of your observation.
Comments:
1.Did the authors observe any sex differences between the Black /AA and their White counterparts in their susceptibility to COVID-19 infections/complications?
Authors’ response: Overall males were more likely to die relative to female. However since the focus the current investigation was on race, we decided not to focus on sex differential in the COVID-19 mortality and complications.
2.Authors should explain the reasons for choosing the states in this study.
Authors Response: Thanks! Very good comment. We have addressed this. The race and ethnicity data were not available on most states, implying the utilization of data from states health department with race and ethnicity data at the time of this modeling.
3.Authors should provide the percentage of populations for Black /AA and their White counterparts in the states they have studied for the COVID-19 pandemic.
Author’s Response: Thanks for the observation, which is necessary in understanding the disproportionate burden of COVID-19 among racial minorities. We used the US Census projection 2020 populations to estimate these burdens. The manuscript has referenced this application.
4.Authors could simplify figure 1, then it will be easier to understand the COVID-19 mortality graphs.
Author’s Response: Thanks for the comment! We do agrere with this observation. However where there are many data points, the visualization allows for the patterns and trends but still hides some visuals. We applied a simplified approach to this figure and simplifying further will imply data reduction by removing some of the states .
5.In table 2, the authors should consider changing the order of Black /AA and White for all the states; the order should be presented uniformly.
Authors Response: Thanks for the observation. We have modified this order.
Reviewer 3 Report
This is such an amazing and timely article and I commend the authors for writing and researching on such an important topic. Here are my edits:
Introduction:
Line 46 – 52: very long sentence, please reduce
Line 57: Please define SDH. SDH fall into 5 categories and it would be important to use them. Look at the CDC, healthy people 2020 and healthy people California for a more updated definition of SDH.
Line 63: low Socioeconomic status (SES) should be low socioeconomic status.
Line 65: Social determinants of health should be SDH since it was already mentioned before
Line 75: define TB as this is the first time you are mentioning it.
Line 78 and 79: give the rate of transmissions and fatality.
Line 81: potential
Line 83: is there supposed to be a full stop between Los Angeles and Race?
Line 87: after defining and abbreviating SDH please make sure to be consistent throughout the manuscript.
Line 80 to 92: the authors only used 1 citation. Please use more references as there are many supporting articles out there to support the claims made in this paragraph. The authors do an excellent job at presenting and setting the foundation for their research, however more active references are needed. Thus far few research studies have been done that can lend support to their argument. For example Yancy (2020) has an excellent article out on COVID-19 and African americans. The CDC also posted a report on April 08th which outlines disproportionate rate of COVID-19 hospitalizations in AA individuals (see Garg et al 2020). There is also a number of articles to support the claim of AA and chronic illnesses like diabetes, hypertension and stroke. More research is needed for this paragraph.
Line 93 – 96: this claim is unsupported – please provide reference.
Line 96-104: Please provide more supporting evidence for these claims.
Line 107: no need to say Blacks and AA are racial minorities – we know and this has already been stated previously. Please provide the reference for your “evidence based data”
Line 112 – 116: too long of a sentence, please shorten. Please support that lifestyle choices are related to structural racism. There Is more than enough evidence in the literature to support such claims. Please provide more than one reference.
Line 118 – 134: I really appreciate how the authors have outlined their case. The foundation is strong and the evidence is presented. However, the authors fail to back their claims with supporting evidence. More references are needed in this section. Maybe some evidence either in this paragraph or the next to show that preliminary blacks are most disproportionally affected in arears like NY, Michigan, Chicago, New Orleans etc.
Line 128: please change “explain causes of causes” this is for causality and experiments: this was no an experiment.
Methods
Line 147: be consistent with headings and subheadings
Line 158: RNA was not defined previously – define if possible
Line 153 Note: this authors used race/ethnicity but consistently referred in the intro as race: confirm race/ethnicity as non-hispanic black/AA. I understand due to the reporting of race and ethnicity that this is hard to make this claim and the researchers did say so which I appreciate. The researchers can also say this in the introduction.
Results
Line 202: New York and California and Lousiana were not mentioned previously but are now in the results and tables. Please state why this is so in either methods or introduction where the first states were mentioned for the analysis. Additionally such states are the epicenters as was rightly stated in the results but not really mentioned in the intro or prior to the results – why?
Line 204 – 235: Please break up this into two or more paragraphs where applicable.
Line 207: again NY and CA among others were not mentioned previously but now included in results
Line222: as stated for line 153: please mentioned earlier that this is about race/ethnicity. In the introduction it was only presented as race.
Figures:
All figures have the name of the figure twice; please be consistent with journal specifications (most journals require this only once – either at the top or bottom of the figure depending on the citation and writing style – not both. I might be wrong as not familiar with journal requirements. Forgive me.
Data was very well presented (2 thumb up). Might be the way the data was uploaded and downloaded. Tables should be on one page and not broken up into two pages as is the case with Table 2 and 3 (might be journal printing error).
Discussion
Line 315: please use urgent and not emergency and urgent. The use of several words to describe one concept leads to run on sentences.
Line 315 – 319: shorten into two or more sentences if possible.
Line328: be consistent with whites vs White
Line 330: Use peer reviewed citations for this as I am sure there is plenty. Garg et al 2020 for example and even state data can provide you with this.
Line 334: REF
Line 341: there is an unneccasary space. I like that the authors mentioned implicit bias, I was wondering why it was not mentioned in the introduction. We cannot talk about this if we do not mention implicit bias. Implicit biases should also be mentioned in the introduction when we talk about structural racism. Black and brown people are turned away with “flu like symptoms” far too often and the pain and symptom management is not taken as serious as if it were white counterparts. See Serena Williams and her birth story. Also mention that Black and Brown ppl are more likely to have jobs and not careers and be essential service and front line workers. In New York City alone, blacks make up at least 75% of all front line workers and immigrants make up 53%. Black people do not have the luxury to stay home. Blacks are also more likely to stay in polluted areas. This is why it is so important to outline the 5 categories of SDH as this is all linked to how black people are suffering endlessly.
Line 330 – 359: ALL great stuff. I am loving it, but please make into to or more paragraphs. Paragraphs should as much as possible have like 9 sentences.
Paragraph beginning at 360. Social determinants of health – replace with SDH.
Line 401 – 445: Please break into smaller paragraphs. So much good information is written in there but it is lost in the cluster of words that make up this very loooooooooooooooooonnnnnnnnnnnnngggggggggg paragraph.
I think that your recommendations make a very valid points. I was hoping to see something on teaching of implicit biases although this is being done in some medical institutions. We have left a lot on the government but as is evidenced by this current pandemic, the one before and perhaps the one after, not much is being done to support the black and brown people and it will continue to be this way unfortunately. One way that we can do this is by having more black and brown physicians. Under-represented minorities in medicine are probably on 4% of physicians. Research shows us that black and brown individuals are more likely to practice in underserved communities where people look and sound like them. When there is patient and physician concordance then there is more likely to be trust.
We also need diversification in research and more funding for research on health disparities and social determinants of health – this is so important … but I is such an uphill climb. Let’s keep publishing and doing the work in this area.
I admire the work that you have undertaken and the recommendations that you have made and I too am hoping for a change.
This was such great work and I thoroughly enjoyed reading this article. Keep spreading the message and hope.
Author Response
Dear Editor and editorial staff:
Thank you and the reviewers for the opportunity to revise our manuscript for a resubmission to your journal. Below please find authors’ response to reviewers’ comments:
Reviewer’s Comments: Line 46 – 52: very long sentence, please reduce
Authors’ Response: addressed, please see the bold.
Line 57: Please define SDH. SDH fall into 5 categories and it would be important to use them. Look at the CDC, healthy people 2020 and healthy people California for a more updated definition of SDH.
Author’s response: Thanks for the comments and suggestions. We did review the CDC notion of SDOH and have addressed this very clearly. Please see the bold..
Reviewer’s comment: Line 63: low Socioeconomic status (SES) should be low socioeconomic status.
Author’s Response: Thanks! We have addressed this.
Line 65: Social determinants of health should be SDH since it was already mentioned before
Author’s Response: Thanks! Addressed.
Line 75: define TB as this is the first time you are mentioning it.
Authors’ Response: Thank you for the observation. Addressed.
Reviewers Comments: Line 78 and 79: give the rate of transmissions and fatality.
Authors response: Thanks for the comments and suggestion. Addressed , please see the bold.
Line 81: potential
Authors Response; Thanks ! Addressed.
Line 83: is there supposed to be a full stop between Los Angeles and Race?
Authors Response: thnajs . No addrtessed.
Line 87: after defining and abbreviating SDH please make sure to be consistent throughout the manuscript.
Reviewers Comment: Line 80 to 92: the authors only used 1 citation. Please use more references as there are many supporting articles out there to support the claims made in this paragraph. The authors do an excellent job at presenting and setting the foundation for their research, however more active references are needed. Thus far few research studies have been done that can lend support to their argument. For example Yancy (2020) has an excellent article out on COVID-19 and African americans. The CDC also posted a report on April 08th which outlines disproportionate rate of COVID-19 hospitalizations in AA individuals (see Garg et al 2020). There is also a number of articles to support the claim of AA and chronic illnesses like diabetes, hypertension and stroke. More research is needed for this paragraph.
Authors Response: Thank you for the comment regarding supporting works in the area . We have provided these references. However it is important to recognize the effort of this team to provide model with larger geographic areas implicated in the disproportionate burden of COVID-19. This work especially with the recommendation is essential for the US public health, governmental initiative and the communities to plan for the second wave by the winter if anticipated.
Reviewers Comments: Line 93 – 96: this claim is unsupported – please provide reference.
Authors response: Thanks for the suggestion to provide supporting data for the information provided herein. We have added a reference to support our position.
Line 96-104: Please provide more supporting evidence for these claims.
Authors Response: We have addesed this. Thanks !
Line 107: no need to say Blacks and AA are racial minorities – we know and this has already been stated previously. Please provide the reference for your “evidence based data”
Authors Response: Thanks! We have addressed this.
Line 112 – 116: too long of a sentence, please shorten. Please support that lifestyle choices are related to structural racism. There Is more than enough evidence in the literature to support such claims. Please provide more than one reference.
Authors Response: Thnaks ! We have provideds an additional data to support
Line 118 – 134: I really appreciate how the authors have outlined their case. The foundation is strong and the evidence is presented. However, the authors fail to back their claims with supporting evidence. More references are needed in this section. Maybe some evidence either in this paragraph or the next to show that preliminary blacks are most disproportionally affected in arears like NY, Michigan, Chicago, New Orleans etc.
Authors Response: Thank you very much for the acknowledgement of the effort of this team. We have provided substantial references to support our stance on these racial variances. Since public health and clinical medicine remain an inaexact science attempts at incorporating previous evidence in this mancsrtit depends on our ability to examine these publish papers for their methodologies prior to citation.
Line 128: please change “explain causes of causes” this is for causality and experiments: this was no an experiment.
Authors Response: We have addressed this….plz see the bold.
Methods
Line 147: be consistent with headings and subheadings
Authors response: addressed.
Line 158: RNA was not defined previously – define if possible
Authors Response: Thanks we have described RNA as ribonucleic acid.
Line 153 Note: this authors used race/ethnicity but consistently referred in the intro as race: confirm race/ethnicity as non-hispanic black/AA. I understand due to the reporting of race and ethnicity that this is hard to make this claim and the researchers did say so which I appreciate. The researchers can also say this in the introduction.
Authors Response: Thanks, we have addressed this.
Results
Line 202: New York and California and Lousiana were not mentioned previously but are now in the results and tables. Please state why this is so in either methods or introduction where the first states were mentioned for the analysis. Additionally such states are the epicenters as was rightly stated in the results but not really mentioned in the intro or prior to the results – why?
Authors Response: Thanks for the observation, which is very accurate. The sates were not mentioned in the into and method section since data were not available at the Health department on race/ethnicity for the entire stsate except in some locale such as st Jonh’s parish in LU. We have addressed these in the intro as well as in the method section.
Line 204 – 235: Please break up this into two or more paragraphs where applicable.
Authors Response: Thanks as ths section was dedicated to the disproportionate burden of COVID-19 based on the population size.
Line 207: again NY and CA among others were not mentioned previously but now included in results
Authors response: Thanks ! We have addressed this.
Line222: as stated for line 153: please mentioned earlier that this is about race/ethnicity. In the introduction it was only presented as race.
Authors Response: Thanks for this observation. We have carefully examined the the entire manuscript and rectified this accordingly.
Figures:
All figures have the name of the figure twice; please be consistent with journal specifications (most journals require this only once – either at the top or bottom of the figure depending on the citation and writing style – not both. I might be wrong as not familiar with journal requirements. Forgive me.
Authors Response: Thanks for this observation, which I do completely agree. We have amended the figures in line with your suggestion and recommendation.
Reviewer’s Comment: Data was very well presented (2 thumb up). Might be the way the data was uploaded and downloaded. Tables should be on one page and not broken up into two pages as is the case with Table 2 and 3 (might be journal printing error).
Authors response: Thanks immensely for this observation. The modelling of aggregate data remains a significant challenge to epidemiologists and biostatisticians. However the knowledge of the underlying cause of the observed public health or clinical medicine phenomenon allows for a reasonable modeling with such data. The data extraction from the referenced public health departments was accurate as well as the analysis.
Discussion
Line 315: please use urgent and not emergency and urgent. The use of several words to describe one concept leads to run on sentences.
Author’s Response: Thanks! We have addressed this.
Line 315 – 319: shorten into two or more sentences if possible.
Authors Response: Thank you very much. We have applied paragraph criteria in generating 2 more paragraphs from the original which has rendered this section very readable. Thanks once again.
Line328: be consistent with whites vs White
Authors Response: Thanks! We have addressed this,
Line 330: Use peer reviewed citations for this as I am sure there is plenty. Garg et al 2020 for example and even state data can provide you with this.
Line 334: REF
Author’s Response: Thanks we have provided a reference as suggested.
Line 341: there is an unneccasary space. I like that the authors mentioned implicit bias, I was wondering why it was not mentioned in the introduction. We cannot talk about this if we do not mention implicit bias. Implicit biases should also be mentioned in the introduction when we talk about structural racism. Black and brown people are turned away with “flu like symptoms” far too often and the pain and symptom management is not taken as serious as if it were white counterparts. See Serena Williams and her birth story. Also mention that Black and Brown ppl are more likely to have jobs and not careers and be essential service and front line workers. In New York City alone, blacks make up at least 75% of all front line workers and immigrants make up 53%. Black people do not have the luxury to stay home. Blacks are also more likely to stay in polluted areas. This is why it is so important to outline the 5 categories of SDH as this is all linked to how black people are suffering endlessly.
Authors Response (Holmes, L Jr) : Thanks ! Very appreciative of these comments. Social determinants of health reflects conditions such as where we live and work. Realistically these SDH interact with human genes and alter gene expression, adversely affecting transcriptomes which are biologic molecules such as drug receptors. When I (Holmes L Jr) lectures on implication of aberrant epigenomic modulations in health disparities, I mention Serena Williams and referenced clinician bias, implicit bias and the clinical notion of “believability” as to why black /AA women are 3 times as likely to die at childbirth and after delivery.
We have applied this in the discussion to reflect the disproportionate burden of transmission and mortality on the communities of color.
Line 330 – 359: ALL great stuff. I am loving it, but please make into to or more paragraphs. Paragraphs should as much as possible have like 9 sentences.
Authors Response: Thanks ! We have addressed this.
Paragraph beginning at 360. Social determinants of health – replace with SDH.
Authors Response: Thanks! We have addressed this.
Line 401 – 445: Please break into smaller paragraphs. So much good information is written in there but it is lost in the cluster of words that make up this very loooooooooooooooooonnnnnnnnnnnnngggggggggg paragraph.
Authors’ Response: Thanks for the observation regarding the length of the paragraph. We completely agree with the comment and had addressed this appropriately.
Reviewers Comments: I think that your recommendations make a very valid points. I was hoping to see something on teaching of implicit biases although this is being done in some medical institutions. We have left a lot on the government but as is evidenced by this current pandemic, the one before and perhaps the one after, not much is being done to support the black and brown people and it will continue to be this way unfortunately. One way that we can do this is by having more black and brown physicians. Under-represented minorities in medicine are probably on 4% of physicians. Research shows us that black and brown individuals are more likely to practice in underserved communities where people look and sound like them. When there is patient and physician concordance then there is more likely to be trust.
We also need diversification in research and more funding for research on health disparities and social determinants of health – this is so important … but I is such an uphill climb. Let’s keep publishing and doing the work in this area.
I admire the work that you have undertaken and the recommendations that you have made and I too am hoping for a change.
This was such great work and I thoroughly enjoyed reading this article. Keep spreading the message and hope.
Authors’ Response: Thank you very much for this observation and comment. We share similar position in addressing disparities as observed in this COVID-19 pandemic. While technology has a key role to play, professional inclusion and diversity is essential in addressing case fatality and mortality racial differentials. We have added recommendation based on your comment. Plz see the bold.
Once again, thank for the opportunity to resubmit our manuscript for a re consideration for publication by your journal.
Very kind regards,
Professor Laurens Holmes,